# Global CO$_2$ uptake of cement in 1930-2019

Rui Guo[1, *], Jiaoyue Wang[2, 3, *], Longfei Bing[2, 3], Dan Tong[4], Philippe Ciais[5], Steven J. Davis[4], Robbie M. Andrew[6], Fengming Xi[2, 3], Zhu Liu[1]

[1] Department of Earth System Science, Tsinghua University, Beijing 100084, China

[2] Institute of Applied Ecology, Chinese Academy of Sciences, Shenyang 110016, China

[3] Key Laboratory of Pollution Ecology and Environmental Engineering, Chinese Academy of Sciences, Shenyang 110016, China

[4] Department of Earth System Science, University of California, Irvine, Irvine, California 92697, USA

[5] Laboratoire des Sciences du Climat et de l'Environnement, CEA-CNRS-UVSQ, CE Orme des 14 Merisiers, 91191 Gif sur Yvette Cedex, France

[6] CICERO Center for International Climate Research, Oslo 0349, Norway

[*] These authors made equal contributions to the work.

*Correspondence to*: Fengming Xi (xifengming@iae.ac.cn) and Zhu Liu (zhuliu@tsinghua.edu.cn)

**Abstract.** Because of the alkaline nature and high calcium content of cements in general, they serve as a CO$_2$ absorbing agent through carbonation processes, resembling silicate weathering in nature. This carbon uptake capacity of cements could abate some of the CO$_2$ emitted during their production. Given the scale of cement production worldwide (4.10 Gt in 2019), a life-cycle assessment is necessary in determining the actual net carbon impacts of this industry. We adopted a comprehensive analytical model to estimate the amount of CO$_2$ that had been absorbed from 1930 to 2019 in four types of cement materials including concrete, mortar, construction waste and cement kiln dust (CKD). Besides, the process CO$_2$ emission during the same period based on the same datasets was also estimated. The results show that 21.02 Gt CO$_2$ (18.01-24.41 Gt CO$_2$, 95% CI) had been absorbed in the cements produced from 1930 to 2019, with the 2019 annual figure mounting up to 0.89 Gt CO$_2$ yr$^{-1}$ (0.76-1.06 Gt CO$_2$, 95% CI). The cumulative uptake is equivalent to approx. 55% of the process emission, based on our estimation. In particular, China's dominant position in cement production/consumption in recent decades also gives rise to its uptake being the greatest with a cumulative sink of 6.21 Gt CO$_2$ (4.59-8.32 Gt CO$_2$, 95% CI) since 1930. Among the four types of cement materials, mortar is estimated to be the greatest contributor (approx. 59%) to the total uptake. Potentially, our cement emission and uptake estimation system can be updated annually and modified when necessary for future low-carbon transitions in the cement industry. All the data described in this study, including the Monte Carlo uncertainty analysis results, are accessible at http://doi.org/10.5281/zenodo.4459729 (Wang et al., 2020).

## 1. Introduction

According to the International Energy Agency (IEA) statistics, cement industry is the second largest industrial $CO_2$ emitter with a share of 27% (2.2 Gt $CO_2$/yr) in 2014 (IEA and WBCSD, 2018) and estimated to account for approximately 7.4% of the total anthropogenic $CO_2$ emission in 2016 (Sanjuán et al., 2020). Broadly, there are two direct sources of $CO_2$ emission originating from cement production: 1. The thermal decomposition of limestone ($CaCO_3$) in the process of producing clinker; 2. the energy required for the decomposition, largely provided by combustion of fossil fuels. For the latter, energy efficiency improvement and cement kiln technology advancement have gained noticeable progress in recent years (Shen et al., 2016; Xu et al., 2014; Zhang et al., 2015). However, it has been widely estimated that the former so-called process emission constitutes most of the total direct emission (approx. 60%). Consequently, the targeted reduction in emission of cement industry for achieving Climate Action SDG[1], which fully aligns with meeting the 'below 1.5°C' climate target (Rogelj et al., 2018), hinges upon reducing process emission. Unfortunately, the traditional standardised Ordinary Portland Cement (OPC), which has been the dominant type of cement used by humans so far, is of very high clinker contents historically i.e. high clinker-to-cement ratio (herein referred as clinker ratio). Both Griffin et al. (1989) and Boden et al. (1995) reported the emission factor (*EF*) to be around 0.5 t $CO_2$/t cement then, which suggested an implicit clinker ratio >95%. On the other hand, since OPC clinkers are CaO-rich, a high clinker ratio would also increase the $CO_2$ absorption capabilities (by carbonation) of cements. The main carbonation mechanisms that are responsible for the carbon uptake of cements can be attributed to their hydroxide(s) and silicate(s) constitutes[2], as described by Eq. (R1) and (R2):

$$Ca(OH)_2 + CO_2 \xrightarrow{\ H_2O\ } CaCO_3 + H_2O \,, \tag{R1}$$

$$Ca_xSi_yO_{(x+2y)} + xCO_2 + zH_2O \rightarrow xCaCO_3 + ySiO_2 \cdot zH_2O \,, \tag{R2}$$

Pan et al. (2020) recently studied the emission reduction potential from producing cement mortar and concrete blocks by mixing in high level of alkaline blending (e.g. blast furnace slag, fly ash and mine tailings) and discovered a yearly multi-giga-tonne potential of $CO_2$ abatement. Therefore, reducing clinker ratio is still the key to lower the process emission level of cement industry while the projected demand for cement is going to increase by 1.1~1.2 times by the end of 2050 (IEA and WBCSD, 2018).

Andrew (2018) updated the global cement industry emission inventory[3] recently by using various data sources for different countries and time periods. The insufficient accounting for the geographically and temporally varying clinker ratio, as was embedded in prior estimation methods adopted by Carbon Dioxide Information Analysis Centre (CDIAC) (Boden et

---

[1] Sustainability Development Goals.

[2] Other minor phases including ettringite also contribute to the overall carbonation (Hyvert et al., 2010).

[3] Process emission only.

al., 2017), was considered and corrected for. On the other hand, in our previous study on the uptake (Xi et al., 2016), clinker ratio values from historical literature, including IPCC (2006) recommended default value of 0.75 (as the lower bound), were used in our model for estimating the uptake as well as the uncertainty analysis by Monte Carlo method. Therefore, updating the results by applying more realistic clinker ratio data is necessary, especially for China where multiple surveys and reports have uncovered the strikingly lower-than-average clinker ratios post-1990.

In this study, we re-estimated the amount of $CO_2$ uptake by cements produced from 1930 to 2019, including those used in concrete and mortar as well as those 'lost' as construction waste and kiln dust. We updated the clinker ratio/production data after 1990 for China and treated India as a separate region. We estimated that 21.02 Gt $CO_2$ (18.01-24.41 Gt $CO_2$, 95% CI) had been absorbed and sequestered in cements that had been produced between 1930 and 2019, which effectively abated 52% of the corresponding process emission. The annual uptake in 2019 alone reached a staggering 0.89 Gt $CO_2$ $yr^{-1}$ (0.76-1.06 Gt $CO_2$ $yr^{-1}$). Using this consistent framework and model, we could include regularly updated annual estimates of cement carbon uptake into annual assessments of the global carbon project (GCP) (Friedlingstein et al., 2019) as an important anthropogenic carbon sink, which has not been thoroughly assessed or documented.

## 2. Data and Methods

### 2.1. Cement/clinker production data resources and treatment

Global cement production data have been estimated by United States Geological Survey (USGS) since 1930s. In our previous study (Xi et al., 2016), we used USGS production data explicitly as the only source for calculations of the uptake. In addition, the world was geographically divided into four primary countries/aggregated regions including China, the United States (US), Europe and Central Eurasia (including Russia) and Rest of the World (ROW). We noticed that, other than Russia and Turkey, the country-specific European and Central Eurasian cement production data was not available yet from USGS after 2017. In this work, to keep the consistency with prior geographical division and data source, 2018 and 2019 cement production data were projected for the 'Europe and Central Eurasia' region. Specifically, the average ratio of the production in Russia and Turkey to the total production in 'Europe and Central Eurasia' from 2013 to 2017 was taken as the scaling factor, so that the total regional production for 2018 and 2019 can be projected assuming this proportion remained the same. For the US, ROW and China (prior to 1990), we continued to use the cement production data since 1930 from USGS. The IPCC recommended clinker ratios were continually used for these aggregated regions without extra fine tuning to country-level data.

In terms of the updates on China, we first collected national cement production data for the period of 1990-2019 from China Statistical Yearbook available from National Bureau of Statistics (NBS, 2019). To calculate the $CO_2$ uptake based on our model (see 2.3 and 2.4), subjecting to data availability for different periods during 1990-2019, we then collected the clinker ratio data from various sources for the 1990-1999, 2000-2014, and 2015-2019 periods were from published literature (Gao et

al., 2017; Xu et al., 2012, 2014), China Cement Almanac (CCA, 2001-2015), and public national data from Ministry of Information Technology (MIIT, 2019), respectively. As such, we also obtained the national clinker production for the 1930-2019 period. Another progress we made in this work was to separate India from the ROW, on the basis that India has now become the second largest cement producer after China with approximately 8% of the world total in 2014 (IEA and WBCSD, 2018). Recent studies (Andrew, 2018, 2020) suggested that the average clinker ratio in India has been fluctuating in the past three decades. Therefore, we used the newly published year-by-year clinker ratio data for India for 1990-2019.

### 2.2. Estimating the process emission

Process $CO_2$ emissions of the cement industry were estimated by multiplying regional clinker production by the derived process $CO_2$ emission factors. Since the process $CO_2$ emissions arise from chemical reactions involved in the production of clinker, as carbonates (largely limestone, $CaCO_3$) are decomposed into oxides (largely lime, $CaO$) and $CO_2$ by the addition of heat, they can be estimated by the conservation of mass flow principle. The default value recommended by IPCC is 510 kg $CO_2$ t$^{-1}$ clinker (Hanle et al., 2006), without considering emissions originating from $MgCO_3$. In this study, we first collected local survey data by kiln type from literature and applied them in the emission estimates. There are mainly five kiln types worldwide, including dry with preheater and precalciner, dry without preheater (long dry kiln), dry with preheater without precalciner, wet/shaft kiln, and semi-wet/semi-dry.

For China, a nationwide sampling survey of 359 cement production lines across 22 provinces was conducted (Shen et al., 2016) and we adopted the process $CO_2$ emission factor estimated from this local Chinese study. As a result, we applied the sample-averaged emission factors: 519.66 kg $CO_2$ t$^{-1}$ clinker for dry with preheater without precalciner, dry with preheater and precalciner, and dry without preheater (long dry) kilns, 499.83 kg $CO_2$ t$^{-1}$ clinker for semi-wet/semi-dry and wet/shaft kilns. For other countries in the absence of detailed survey data, we adopted the emission factors that were collected and summarised in (Andrew, 2018), which integrated local emission information for key countries (e.g. India). We then obtained annual country- or regional-level production technology information from the World Business Council for Sustainable Development (WBCSD) and the Global Cement Directory 2019 (publicly named as the GCD-2019 dataset). While WBCSD collected technology-based clinker production information using a survey-based approach (IEA and WBCSD, 2018), the GCD-2019 dataset provides plant-level information of cement industries in service as of 2019 (for example, cement production capacity, physical address, number of kilns, and cement production technology etc.). We then cross-checked and integrated the 'start of operation year' information at plant level from the 'industryAbout' database (industryAbout, 2019) and various companies' websites. This information enabled us to infer the annual capacity-weighted production technology (i.e. kiln types) distributions for the 1930-2019 period. Finally, we used technology-weighted emission factors to calculate the regional average emission factors, which were then used to estimate process $CO_2$ emissions directly.

It is noted that in order to stay in line with the life-cycle $CO_2$ uptake assessments of concrete structures, concrete construction waste and CKD in this study, in comparison to some previous studies (e.g. Andrew, 2018), our estimation framework for

process $CO_2$ emissions is relatively simple. Nevertheless, we integrated the global plant-level capacity and technology information into our estimates for the first time, to provide new perspectives on emission estimates. Besides, we also assessed the uncertainties of such estimates using Montel Carlo method.

### 2.3. Life-cycle uptake assessments of concrete structures

Here, we adhere to the breakdown of concrete utilisation into 3 stages as before (Xi et al., 2016): 1. Service; 2. Demolition; 3. Secondary use. Therefore, the carbon uptake of concrete ($C_{con}$) can be calculated as an aggregate of the three subcomponents:

$$C_{con} = C_{l,tl} + C_{d,td} + C_{s,ts},$$  (1)

where $C_{l,tl}$, $C_{d,td}$ and $C_{s,ts}$ are the uptake during service, demolition and secondary use stage, respectively. The life cycle was deemed to be 100 years in line with a historical study by Pade and Guimaraes (2007), considering the longest average life of buildings in Europe is merely 70 years (Pommer and Pade, 2005). During concretes' service life, they are used primarily to build various functional buildings, roads, utilities and other public works etc., hence exhibiting different sizes and geometrical shapes in the environment. We adopt a simplified approach by considering a three-dimensional diffusion 'slab' model in which carbonation starts at the exterior side of the slab and gradually moves inwards: this is schematically shown in Figure 1. According to Fick's 2$^{nd}$ Law, which is used in the calculation[4], the carbonation depth is proportional to the square root of the carbonation time ($t_l$) i.e. $d_i = k_i \sqrt{t_l}$ linked by an apparent diffusion coefficient ($k_i$). According to Eq. (R1) and (R2), in order to determine the amount of $CO_2$ being absorbed during the carbonation processes, it is pivotal to work out the amount of Ca cations in the cements i.e. 1 mol Ca cation takes 1 mol of $CO_2$. Similar to what was recommended by the IPCC regarding the calculation of cement emission factor[5], the theoretical carbon uptake of cements also depends on the clinker ratio ($f_{cem}^{clinker}$) and on the CaO content in the clinker ($f_{clinker}^{CaO}$). Additionally, in natural conditions, not all of the calcium in OPC would be associated with carbonation reactions due to its microstructural constraints (Lagerblad, 2005) hence the fraction of CaO that could be converted to $CaCO_3$ ($\gamma$) should be considered, too, as follows:

$$C_{cem} = f_{cem}^{clinker} f_{clinker}^{CaO} \gamma \frac{M_{CO_2}}{M_{CaO}},$$  (2)

where $M_{CO_2}$ and $M_{CaO}$ are the molar mass of $CO_2$ and CaO, respectively.

---

[4] Other more sophisticated diffusion models have not been widely accepted or verified.

[5] $E_{cem} = f_{cem}^{clinker} f_{clinker}^{CaO} \frac{M_{CO_2}}{M_{CaO}}$.

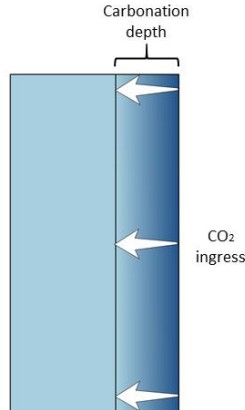

**Figure 1 A two-dimensional schematic representation (rectangular cross section) of the three-dimensional 'slab' carbonation model of concretes. The right-hand side that is close to the $CO_2$ source is being carbonated first with further carbonation takes place by $CO_2$ diffusion in the cement.**

In order to estimate the carbon uptake at macroscopic scale with the data available, we made the following simplifications:
1. Assuming the diffusion front is equivalent to the carbonation front and the area behind the front is regarded fully carbonated[6];
2. Assuming the geometries of the cement parts resemble the slab shown in Figure 1 so that the exposed surface area ( $A_i$ ) can be calculated by the concrete volume in different structure categories and average thickness data. Further, since the carbonation rate depends on the environmental conditions e.g. humidity and temperature, $CO_2$ concentration etc. and the concrete's physiochemical conditions e.g. compressive strength, additives and surfacing etc., we further broke down the utilisation of concrete based on these specifications (see http://doi.org/10.5281/zenodo.4459729 (Wang et al., 2020)). The regional-specific calculations were then realised by regrouping the data based on their region-specific sources. Consequently, the regional and global uptakes can be calculated by aggregating each compressive strength class ( $i$ )[7] as

$$C_l^{tt} = (\sum_i d_i * A_i * c_i) f_{cem}^{clinker} f_{clinker}^{CaO} \gamma \frac{M_{CO_2}}{M_{CaO}}, \tag{3}$$

where the common symbols keep their meanings as defined previously and $c_i$ stands for the cement content of concrete. In short, on top of the regional cement production and/or clinker ratio data, other statistics necessary to carry out such regional calculations include (all regional) the proportion of cement used for making concrete (as opposed to mortar), the cement contents, the CaO content of clinker, the distribution of compressive strength class and the average thickness of different

---

[6] As opposed to the concept 'partly-carbonated', where reaction kinetics are considered.

[7] Four class strengths are considered including ≤C15, C16-C23, C24-C35, >C35.

concrete utilisations. Crucially though, diffusion coefficients of $CO_2$ in concretes of the above specifications and the corresponding service lives will dictate how rapidly and for how long the uptake lasts. We conducted an extensive literature survey to collect the data needed on a regional basis and used collected datasets representative of Europe for ROW, apart from the concrete utilisation data, which we opted to apply the Chinese situation[8] and service lives, which we derived directly from literature (see http://doi.org/10.5281/zenodo.4459729 (Wang et al., 2020)).

After their service life, concretes are usually demolished for either landfill or being reused. Reusing concrete at the end of its service life has been encouraged and envisaged to reduce the total emissions and increase the sustainability of the cement industry (IEA and WBCSD, 2018). However, the reusing rate of demolished concrete had been found to be very low at about 25% worldwide (Kikuchi and Kuroda, 2011; Yang et al., 2014). Demolition entails crushing of the bulk concrete structures so that the embedded steel structures can be easily extracted and recycled hence the end-product is usually in broken piece. Therefore, the surface area exposed to the air dramatically increase during the demolition stage. As pointed out earlier (Eq. (3)), the exposed surface area is one of the key parameters that is positively correlated with the rate of carbonation, it is therefore expected that the carbon sequestered per unit time would increase with an increasing exposure. Again, we simplified the geometrical aspects of the calculations by assuming the demolished and crushed concrete parts ended up in spherical shapes so that the carbonation starts from the outer surface moving inwards radially (see Figure 2). Similarly, we considered the same diffusion model to be applied for the carbonation process. Based on the survey of typical crushed cement particle sizes, we divided the distributions into three distinct groups according to their respective minimum (a) and maximum diameters (b) in the range, with respect to the maximum diameter ($D_{0i}$) that a particle will undergo full carbonation in compressive strength class $i$: 1. b $\leq D_{0i}$; 2. a $\leq D_{0i} <$ b; 3. a $> D_{0i}$. The corresponding methods for calculating their carbonated fraction ($F_{di}$) then are as follows:

$$
F_{di} = \begin{cases}
1 - \int_a^b \dfrac{\pi}{6}\left(D - D_{0i}\right)^3 \Big/ \int_a^b \dfrac{\pi}{6} D^3 & (a > D_{0i}) \\[3ex]
1 - \int_{D_{0i}}^b \dfrac{\pi}{6}\left(D - D_{0i}\right)^3 \Big/ \int_a^b \dfrac{\pi}{6} D^3 & (a \leq D_{0i} < b) \\[3ex]
1 & (b \leq D_{0i})
\end{cases}
$$

---

[8] The rationale is that ROW is mainly comprised of developing nations, hence it is more likely that the utilisation of concrete adopts similar patterns to China.

$$D_{0i} = 2d_{di} = 2k_{di}\sqrt{t_d} ,$$ (4)

where $k_{di}$ and $t_d$ [9] are the diffusion coefficient of 'exposed to air' condition for compressive strength class $I$ and the time between service life and subsequent dealings. Besides, based on the survey data from literature for the particle size, we assumed a uniform distribution between a and b for each reginal subcategory.

Since carbonation during the demolition stage took place only in the bulk of concrete material where it remains uncarbonated after used in service, the fraction of carbonated concrete before demolition should be excluded from the calculation to avoid double counting. We assigned the total mass of consumed cement as $m_{ci}$ [10] and the carbonated cement in service life as $m_{li}$ ($m_{li} = d_i * A_i * c_i$ as in Eq. (3)). Therefore, the total amount of $CO_2$ uptake during the demolition stage ( $C_d^{td}$ ) can be calculated as:

$$C_d^{td} = \sum_i (m_{ci} - m_{li}) F_{di} f_{cem}^{clinker} f_{clinker}^{CaO} \gamma \frac{M_{CO_2}}{M_{CaO}} ,$$ (5)

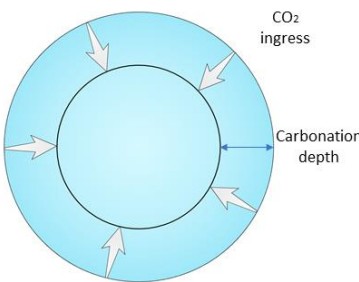

**Figure 2 Two-dimensional schematic representation (circular cross section) of the three-dimensional 'sphere' carbonation model of a concrete particle in the demolition stage.**

Carbonation during the secondary use stage that follows would be slower because a carbonate layer has formed at the particle surface previously. It might be less confusing to the readers to think of the demolition and secondary use stages as a whole with the diffusion process slowing down during the latter. Additionally, because of the high rates of landfill post-demolition, the diffusion processes are further retarded in the buried conditions [11] (Papadakis et al., 1991; Yoon et al., 2007). Therefore, we introduce a lag time $\Delta t$ for it would take longer for the carbonation to reach the same depth ( $d_{di}$ ) when concrete

---

[9] The average value was estimated to be 0.4 years worldwide (Pade and Guimaraes, 2007).

[10] The cement consumed was taken the same as the cement produced. The discrepancies were sought after for certain years and considered in the uncertainty analysis.

[11] Demolished concretes that are subsequently landfilled and recycled as backfill aggregates are assumed devoid of further carbonation.

particles are in the secondary use conditions compared with the demolition conditions ($t_{di}$):

$$d_{di} = k_{di}\sqrt{t_{di}} = k_{si}\sqrt{t_{di} + \Delta t} \; , \tag{6}$$

and we have:


$$\Delta t = t_{di}\left(\left(\frac{k_{di}}{k_{si}}\right)^2 - 1\right), \tag{7}$$

where the common symbols shared with Eq. (4) have the same meanings and $k_{si}$ stands for the diffusion coefficient during secondary use for compressive strength class $i$. By now, we can represent the combined carbonation depth of demolition and secondary use stages with all known variables as:

$$d_{ti} = k_{si}\sqrt{t_{di} + \Delta t + t_{si}} \;\; (D_{ti} = 2d_{ti}), \tag{8}$$

where $t_{si}$ is the average time of the secondary use stage[12] and $D_{ti}$ is the maximum diameter that a particle will undergo full carbonation in compressive strength class $i$ in the demolition and secondary use stages combined. Similar to how we determined the fraction of carbonation previously, the fraction of further carbonation during the secondary use stage can be calculated by integration according to the same set of particle size criteria:

$$F_{si} = \begin{cases} 1 - \int_a^b \frac{\pi}{6}\left(D - D_{ti}\right)^3 \Big/ \int_a^b \frac{\pi}{6} D^3 - F_{di} & (a > D_{ti}) \\ \\ 1 - \int_{D_{ti}}^b \frac{\pi}{6}\left(D - D_{ti}\right)^3 \Big/ \int_a^b \frac{\pi}{6} D^3 - F_{di} & (a \le D_{ti} < b) \\ \\ 1 & (b \le D_{ti}) \end{cases}, \tag{9}$$


      Like for the demolition stage where double-counting was avoided by excluding the carbonated concrete during service, calculating the carbonation during secondary use should be based on the uncarbonated fraction of concrete after the service and demolition stages. Accordingly, the carbon uptake during the last piece of the concrete life cycle can be expressed as follows:

$$C_s^{ts} = \sum_i (m_{ci} - m_{li} - m_{di}) F_{si} \, f_{cement}^{clinker} \, f_{clinker}^{CaO} \, \gamma \frac{M_{CO_2}}{M_{CaO}}, \tag{10}$$

where $m_{di}$ stands for the mass of concrete carbonated during the demolition stage. Overall, Eq. (1) can be applied to obtain

---

[12] Since the life cycle of concrete is assessed on a 100-year basis, $t_{si} = 100 - t_{li} - t_{di}$.

the total amount of $CO_2$ absorbed based on the three stages as outlined above.


### 2.4. Kinetic uptake models for other types of cement materials

#### 2.4.1. Carbon uptake of cement mortar structures

Here, we adhere to the breakdown of mortar utilisation into 3 sub-components as before (Winter and Plank, 2007; Xi et al., 2016): 1. Rendering and plastering mortar; 2. Masonry mortar; 3. Maintenance and repairing mortar. Therefore, the total

carbon uptake of mortar ($C_{mor}$) can be calculated as an aggregate of the three sub-components:

$$C_{mor} = C_{rpt} + C_{rmt} + C_{rmat},$$ (11)

where $C_{rpt}$, $C_{rmt}$, and $C_{rmat}$ are the uptake of the corresponding component, respectively. For each sub-component, we conducted an extensive literature survey to collect the Chinese cement mortar utilisation category and percentage data. Additionally, since mortar carbonisation has not been quantified before as far as we are aware, we conducted experiments to

measure the mortar carbonation rate coefficients and the proportion of CaO converted to $CaCO_3$ of typical mortar cements produced in China and used these measured datasets as being representative of the other regions owing to a lack of data. Like concrete, mortar carbonation processes were also simplified to a two-dimensional diffusion 'slab' model in which carbonation starts at the exterior of the slab and gradually moves inwards, and similarly, Fick's 2$^{nd}$ Law was applied to determine the carbonation depth in the general form. However, mortar cement diffusion rates ($K_m$) were shown to be higher than concrete

which has a lower cement content, higher water/cement ratios, and finer aggregate grains (El-Turki et al., 2009). The total mortar carbonation can be determined based on Eq. (2) and (11) with the corresponding proportion of CaO conversion ($\gamma_1$, see Eq. (14)) adjusted to the mortar situation as measured. Again, we assumed the diffusion front is equivalent to the carbonation front and the area behind the front was regarded fully carbonated.

The large exposure area and thin layers of mortar cement translate into rapid carbonation. We calculate annual mortar

cement carbon uptake based on the proportion of annual carbonation depths of the utilisation thicknesses. The annual carbonation of mortar used for rendering, plastering, and decorating is calculated as follows:

$$d_{rp} = K_m \times \sqrt{t} \, , \tag{12}$$

$$f_{rpt} = \left( d_{rpt} - d_{rp(t-1)} \right) / d_{Trp} \times 100\% \, , \tag{13}$$

$$C_{rpt} = W_m \times r_{rp} \times f_{rpt} \times f_{cem}^{clinker} \times f_{clinker}^{CaO} \times \gamma_1 \times \frac{M_{CO_2}}{M_{CaO}} \, , \tag{14}$$


where $d_{rp}$ is the carbonation depth of rendering mortar, $K_m$ is the carbonation rate coefficient of cement mortar, $t$ is the exposure time of rendering mortar after construction, $f_{rpt}$ is the annual carbonation percentage of cement used for rendering mortar in year $t$, $d_{rpt}$ and $d_{rp(t-1)}$ are the carbonation depths of rendering mortar in year $t$ and $(t-1)$, respectively, $d_{Trp}$ is the utilisation thickness of rendering mortar, $C_{rpt}$ is the annual carbon uptake of carbonated rendering mortar, $W_m$ is the amount

of cement for producing mortar, $r_{rp}$ is the percentage of rendering mortar cement of total mortar cement, $\gamma_1$ is the proportion of CaO within fully carbonated mortar cement that converts to $CaCO_3$. After the carbonation depths in adjacent years were determined, the annual carbonation percentage was obtained by the difference between adjacent years to the total utilisation thickness. Combined with the cement for mortar and the percentage of mortar for rendering survey data, the annual carbonation of rendering mortar is then quantified. Calculation for carbon uptake of repairing and maintaining cement mortar is similar to

rendering, plastering, and decorating mortar, with differences lie in the utilisation thickness and the percentage of mortar for repairing and maintaining.

In comparison to mortars for rendering and repairing, it takes longer for masonry mortar to complete carbonation due to the partially exposed condition, thicker utilisation layers and their covering by rendering mortar on masonry wall surfaces. Here, we classify masonry walls into walls with both sides rendered ($C_{mbt}$), walls with one side rendered ($C_{mot}$), and walls

without rendering ($C_{mnt}$). We conducted an extensive survey to collect data on the extents to which mortar rendering has been applied to masonry walls in China and used the representative data of China for other regions due to a lack of data. The carbon uptake of masonry mortar can be calculated as an aggregate of the three subcomponents:

$$C_{rmat} = C_{mbt} + C_{mot} + C_{mnt} \, , \tag{15}$$

where $C_{mbt}$, $C_{mot}$, and $C_{mnt}$ are the uptake of the corresponding classification, respectively. The schematics of the carbonation

models for the three situations mentioned are shown in Figure 3 i.e. carbonation starts from the exterior rendering layer into

masonry layer when having one- and two-side rendering layer or directly from the masonry layer and gradually moves inwards when without rendering.

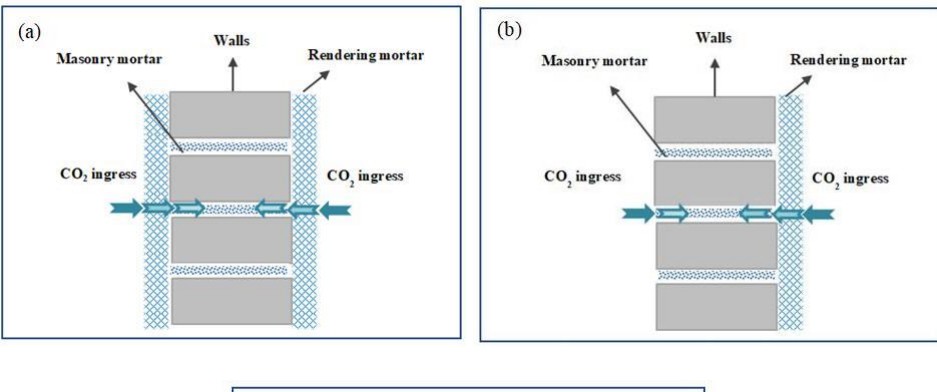

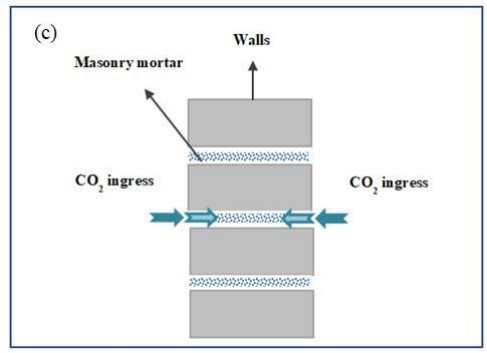

**Figure 3 Two-dimensional schematic representations (cross section) of the carbonation model for masonry mortar. (a) Masonry mortar with one-side rendering carbonation takes place by $CO_2$ diffusion from both rendering layers inwards masonry; (b) Masonry mortar with one-side rendering carbonation takes place by $CO_2$ diffusion from the render layer on one side and directly from masonry on the other; (c) Masonry mortar without render carbonation takes place by $CO_2$ diffusion directly from the exterior of masonry.**

Based on the models outlined above, the calculation of masonry mortar carbonation is similar to rendering and repairing mortar in that determining the annual carbonation is according to the proportion of carbonation depth. The carbonation of masonry mortar for walls with both sides rendered is as follows:

$$d_{mb} = \begin{cases} 0 & (t \le t_r) \\ 2\left(K_m \times \sqrt{t} - d_{Trp}\right) & (t > t_r) \end{cases} , \tag{16}$$

$$f_{mbt} = \begin{cases} 0 & (t \le t_r) \\ \left(d_{mbt} - d_{mb(t-1)}\right)/d_w \times 100\% & (t_r < t \le t_{sl}) \\ 100\% - d_{mbt_{sl}}/d_w \times 100\% & (t = t_{sl} + 1) \end{cases} , \tag{17}$$


$$C_{mbt} = W_m \times r_{rm} \times r_b \times f_{mbt} \times f_{cem}^{clinker} \times f_{clinker}^{CaO} \times \gamma_1 \times \frac{M_{CO_2}}{M_{CaO}}, \tag{18}$$

where $d_{mb}$ is the total carbonation depth of masonry mortar of wall with both sides rendered, $t$ is the exposure time of masonry mortar after construction, $t_r$ is the time of full carbonation of rendering mortar of thickness $d_{Trp}$ and $d_{Trp}$ is the thickness of rendering mortar on masonry wall, $f_{mbt}$ is the annual carbonation percentage of cement used for masonry mortar with both sides rendered in year $t$, $d_{mbt}$ and $d_{mb(t-1)}$ are carbonation depth of masonry mortar with both sides rendered in year $t$ and ($t$-

1), respectively, $d_w$ is the thickness of masonry wall, $t_{sl}$ is the building service life, $d_{mbt_{sl}}$ is the carbonation depth of a masonry mortar with both sides render in building service life, $C_{mbt}$ is the annual carbon uptake of cement for masonry mortar with both sides rendered in year $t$, $r_{rm}$ is the percentage of masonry mortar cement in total mortar cement, $r_b$ is the percentage of masonry mortar with both sides rendered of total masonry mortar. Masonry mortar for walls with both sides first start carbonation from both sides of exterior rendering and then gradually moves inward. When the utilisation time of masonry

mortar is shorter than the time required for full carbonation of rendering mortar on the masonry wall, there is no chance for the underlying masonry to be exposed to $CO_2$, hence carbonation should not happen; otherwise, the fraction of carbonated rendering mortar on the surface should be excluded from the calculation to avoid double-counting. If the utilisation time of masonry mortar is longer than the time required for full carbonation of rendering mortar but shorter than the building service life, we used the carbonation depth difference between adjacent years to the total thickness to show the carbonation fraction

of masonry mortar with both sides rendered in year $t$. If the utilisation time of masonry mortar is longer than the building service life, it was assumed that the left uncarbonized masonry mortar will be fully carbonated in one year due to the large exposure area post-demolition. Therefore, the fraction of masonry mortar carbonation after service life can be quantified by the difference between the fraction of masonry mortar carbonation in service life (that is, the carbonation depth during the service life to the total thickness) and the total masonry mortar carbonation of 100%. The calculation for the carbonation of

masonry mortar for walls with one side rendered differ only in the carbonation depth calculation i.e. without rendering on one side, $CO_2$ directly contacts the bare masonry mortar, so that only the fraction of carbonated rendering mortar on one side was

excluded. Similarly, for walls without rendering at all, $CO_2$ directly contacts the bare masonry mortar from both sides so that the total carbonation depth is twice of the carbonation depth on one side.

### 2.4.2. Uptake assessments of construction wastes

Cement wastes mainly arise during construction and accounts for 1% to 3% of total cement consumption according to construction budget standards (Zhou, 2003) and survey data (Lu et al., 2011). Most of this waste is in small pieces and will be recycled as back fill or landfilled after the completion of building projects, of these about 45% is concrete and 55% is mortar (Bossink and Brouwers, 1996; Huang et al., 2013). Here, we adhere to the breakdown of construction wastes into 2 components as before: 1. Construction waste mortar; 2. Construction waste concrete. The carbon uptake of construction wastes ( $C_{waste}$ ) then can be calculated as an aggregate of the two subcomponents:

$$C_{waste} = C_{wastecon} + C_{wastemor} ,$$
(19)

where $C_{wastecon}$ and $C_{wastemor}$ are the uptake of the corresponding component, respectively. Given the small piece sizes and hence large exposure area of the construction wastes, we made a few simplifications according to the literature survey: 1. Assuming waste mortar completely carbonate in the first year; 2. Assuming waste concrete completely carbonate over the following 5 years (ranging from 1 to 10 years) (Bossink and Brouwers, 1996; Huang et al., 2013). Consequently, the carbon uptake of construction wastes can be quantified by the annual carbonation fraction in line with the ratio of carbonation depths to the full carbonation depths. The expression of construction wastes carbonation as following:

$$C_{wastecon} = (\sum_1^n W_{ci} \times f_{con} \times r_{cont}) \times f_{cem}^{clinker} \times f_{clinker}^{CaO} \times \gamma \times \frac{M_{CO_2}}{M_{CaO}} ,$$
(20)

$$C_{wastemor} = (\sum_1^n W_{mi} \times f_{mor} \times r_{mor}) \times f_{cem}^{clinker} \times f_{clinker}^{CaO} \times \gamma_1 \times \frac{M_{CO_2}}{M_{CaO}} ,$$
(21)

where $W_{ci}$ is the cement used for concrete in strength class $i$; $f_{con}$ is the loss rate of cement for concrete in construction stage, $r_{cont}$ is the annual carbonation fraction of construction waste concrete, $W_{mi}$ is the cement used for mortar in strength class $i$, $f_{mor}$ is the loss rate of cement for mortar, $r_{mor}$ is the annual carbonation fraction of construction waste mortar. In short, in addition to the regional cement production, clinker ratio data, other statistics needed to conduct the calculation include the distribution of compressive strength class, the loss rate of cement for concrete and mortar in construction stage as well as the carbonation time of construction wastes. Crucially though, the latter two statistics, for which we collected the data on a regional basis, will dictate the amount of carbon uptake.

### 2.4.3. Uptake assessments of cement kiln dust

Cement kiln dust (CKD) is the major by-product of cement manufacturing process and has traditionally been considered as an industrial waste. Most of CKD is diverted to landfills and a small part is beneficially reused (Khanna, 2009; USEPA, 1993). CKD is composed of fine, powdery solids and highly alkaline particulate material, and is similar in appearance to Portland cement (Seo et al., 2019). Given the very small particle size (predominantly ranges from a few microns to 50 μm and some coarse particles between 50–100 μm, (Kaliyavaradhan et al., 2020)), CKD full carbonation in landfill conditions can be achieved very rapidly within one year and indeed substantial carbonation even occurs within the first 2 days of reaction (Huntzinger et al., 2009a, 2009b; Siriwardena and Peethamparan, 2015). Therefore, the carbon uptake by CKD is calculated as follows:

$$C_{CKD} = W_{cem} \times f_{cem}^{clinker} \times r_{CKD} \times r_{landfill} \times f_{CKD}^{CaO} \times \gamma_2 \times \frac{M_{CO_2}}{M_{CaO}}, \tag{22}$$

where $W_{cem}$ is the cement production, $r_{CKD}$ is the CKD generation rate based on clinker production, $r_{landfill}$ is the proportion of CKD treatment in landfill, $f_{CKD}^{CaO}$ is the CaO proportion in CKD (Siriwardena and Peethamparan, 2015), $\gamma_2$ is the fraction of CaO within fully carbonated CKD that has been converted to $CaCO_3$. This equation stands because of the fact that CKD carbonation effectively completes within one year.

### 2.5. Yearly and cumulative uptake calculations

While the sectoral carbon uptake can be analytically estimated by the corresponding sectoral equations i.e. for concrete, mortar, construction waste and CKD, respectively, using aggregated regional datasets as the inputs, the regional carbon uptake was determined by aggregating all sectoral contributions but with disaggregated regional production/consumption and diffusion/carbonation coefficient, concrete structure thickness, concrete strength distribution, mortar utilisation distribution, waste particle distribution and CKD generating rate data, among others, as the model inputs. Consequently, the world total uptake can be divided up according to the usage of the cement produced as well as where the cement was produced/consumed[13].

For mortar cement, we explicitly showed how to determine the annual carbonation from Eq. (12) to (14) and Eq. (16) to (18). Basically, for the carbon uptake of a specific year $t$, we can apply a simple subtraction of the cumulative values between adjacent years as:

---

[13] The discrepancy between production and consumption was included in the uncertainty analysis.

$$\sum_j C_t = C_{cem,j}^t - C_{cem,j}^{t-1} \ (j = con, mor, waste, CKD)\text{,}$$

$$\tag{23}$$

so that each year's contribution to the total carbonation can also quantified. This way, we will be able to visualise the time lag in the carbonation process in that the uptake of a specific year $t$ is not limited to the cement produced in the same year.


### 2.6. Uncertainty analysis

Based on the kinetic models described in previous sections, the annual regional carbon uptake was calculated by aggregating the contributions from individual types of cement. Likewise, the annual global carbon uptake was obtained from regional aggregation. It should be noted, though, that a Monte Carlo analysis method with 26 variables (see
http://doi.org/10.5281/zenodo.4459729 (Wang et al., 2020)) was applied to evaluate the carbon uptake at each level, hence the annual median at a higher level (i.e. regional wrt. cement type and global wrt. regional) is not equal to the sum of its sublevel components. The variables associated with the estimates are mostly in common with our previous study (Xi et al., 2016), with the only difference being the distribution of the clinker ratio. Previously, the clinker ratio was set to range from 75% to 97%, in a Weibull distribution with shape and scale parameters of 91.0% and 25 for the years of 1990-2019. In this research, for
China, based on previous studies and local survey data, we adjusted the corresponding uncertainty range for the 1990-2019 period.  Specifically, for 1990-2004, the range of coefficient value of clinker ratio was set to 10%-20%. In this range, the pseudo-random numbers were generated with a uniform distribution then multiplied by the mean values of clinker ratio to obtain the corresponding standard deviation. As such, the normal distributed random clinker ratio values were created. For 2004-2019, the random errors were calculated within the range of ±5% of the mean values with a uniform distribution. For
1930 -1989, the clinker ratio distribution was unchanged.

On the other hand, emission estimates are subject to uncertainties due to incomplete knowledge of activity levels and emission factors. In order to assess the uncertainties in our results more thoroughly, we conducted a comprehensive analysis of regional emission estimates. Following the method of previous studies (Tong et al., 2018; Zhao et al., 2011), we performed a Monte Carlo analysis that varied key parameters including cement production, clinker ratio, and emission factors. The term
"uncertainty" in this study refers to the lower and upper bounds of a 95 % confidence interval (CI) around our central estimate i.e. median. All of the input parameters of activity levels and emission factors, with corresponding statistical distributions, were fed into a Monte Carlo framework and 10,000 simulations were performed to analyse the uncertainties of estimated $CO_2$ emissions. For the uncertainties of the regional process $CO_2$ emission estimates, national average emission factors were derived from previous studies and local survey databases (Andrew, 2018; Hanle et al., 2006; Shen et al., 2016) and we assumed these
activity rates are normally distributed, with coefficients of variations (CV i.e. the standard deviation divided by the mean) ranging from 0.05 to 0.2 based on the specific data sources and year. Furthermore, the ranges of parameter values also vary by country in part due to the quality of their statistical infrastructure.

### 3. Results and Discussions

#### 3.1. Aggregated regional and global process emission

With the continued increase in the production of cement and associated clinker globally in the past few decades, the process $CO_2$ emissions correspondingly have been increasing with limited abating measures (i.e. carbon capture and storage, CCS). According to our estimates, by 2019, the global process $CO_2$ emissions reached 1.57 Gt yr$^{-1}$ (1.42 Gt-1.86 Gt, 95% CI) (see Figure 4a), equivalent to about 25% of the total $CO_2$ emissions from industrial activities in 2018 (Tong et al., 2019). Cumulative emissions from 1930 to 2019 were estimated to be 38.22 Gt (95% CI, 36.98 Gt-40.06 Gt), and more strikingly, more than 71% of which have occurred since 1990. This finding agrees with other studies on cement carbon uptake using similar modelling approaches (Cao et al., 2020). From 1930 to 2019, with the rapid increase of cement demand (+5777% of cement production increase during 1930-2019) driven by global industrialisation and urbanisation, the total process $CO_2$ emissions correspondingly increased by about 49 times, which is actually slightly slower than the increase in production. This is partly due to the relative decreases in average clinker ratios (from ~89% in 1930 to ~70% in 2019) (Wang et al., 2020). Meanwhile, the regional attribution of such an increase changed significantly during the same period. As we can see in Figure 4b, the process emission from cement produced in China and ROW (mainly comprising developing countries) gradually replaced the dominant roles of the US and Europe (and central Eurasia) while there has been a considerable growth for India after 2000s, contributing more than 85% to the total emissions (1.42 Gt, 95% CI, 1.34 Gt -1.59 Gt) in 2019 altogether. Specifically, China alone emitted more than half (~53%, 0.75 Gt, 95% CI, 0.69 Gt -0.89 Gt) and India, as the second largest cement production country in the world, emitted ~10% of the total process emissions of the cement industry as of 2019.

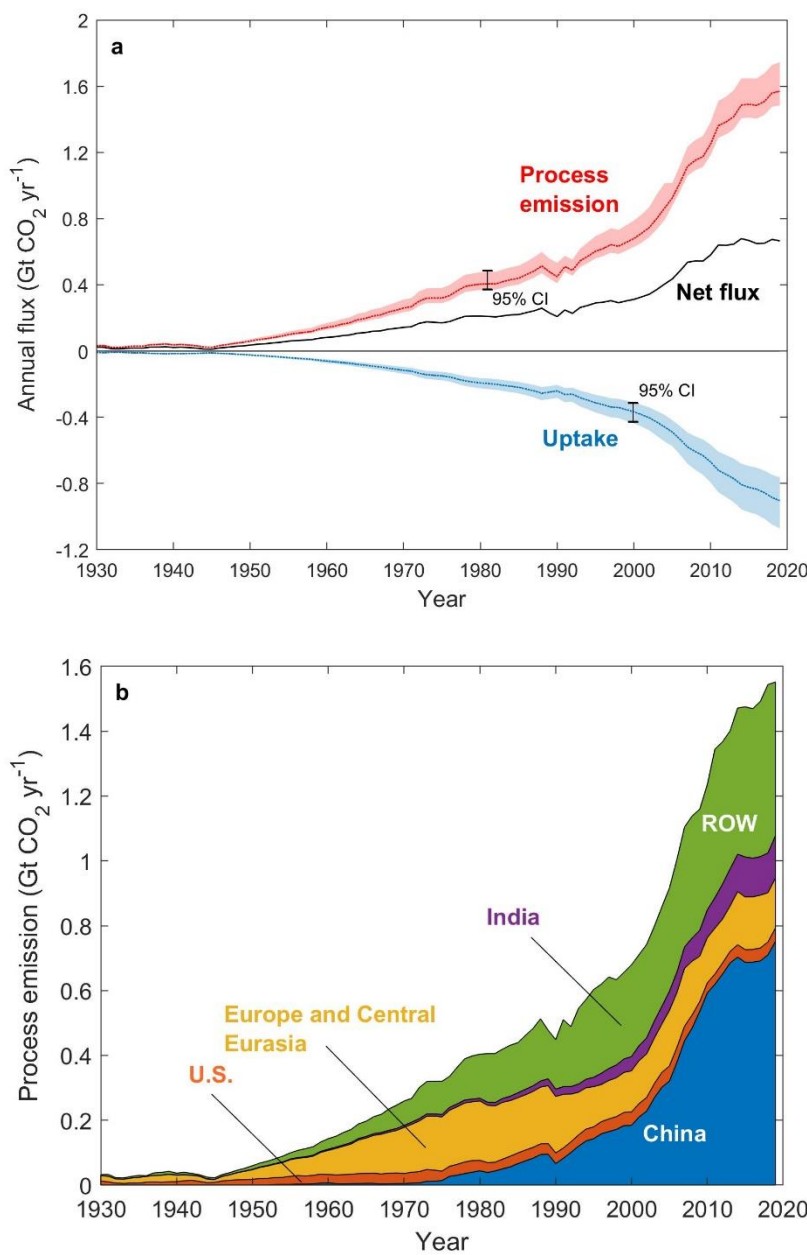

**Figure 4 a) Annual CO₂ process emission and uptake from producing and utilising cement materials from 1930 to 2019. The dotted lines denote the median values while the respective shaded area denote the 95% confidence interval from Monte Carlo simulations. The net emission is also illustrated with the solid black line; b) The country/region-wise process CO₂ emission (median) from 1930 to 2019.**

As mentioned in 2.2, there are other studies estimating the process emission based on high-resolution, national-level

clinker ratio data. Andrew (2019) reported the process emission in 2017 to be $1.48 \pm 0.20$ Gt $CO_2$ and that of aggregated 1928-2017 to be $36.9 \pm 2.3$ Gt $CO_2$. Using our simpler regional based approach gives the 2017 process emission to be 1.37 Gt $CO_2$ (1.30-1.53 Gt $CO_2$, 95% CI) while the 1930-2017 cumulative process emission to be 35.3 Gt $CO_2$ (32.8-40.5 Gt $CO_2$, 95% CI). The results are very similar and unsurprisingly our estimates have a greater level of uncertainty due to our coarser disaggregation of geographic regions.

### 3.2. Cement carbon uptake by region and material type

Global $CO_2$ uptake by cement materials in 2019 reached 0.89 Gt (0.76~1.06 Gt, 95% CI) according to our estimates, of which cement consumed in China contributed about 0.40 Gt. Cumulatively speaking, China, as a country, also made the greatest contribution mounting up to 6.21 Gt $CO_2$ (4.59-8.32 Gt $CO_2$, 95% CI). This is clearly illustrated in Figure 5a where the area representing each region denotes the amount of uptake. In the US and Europe, since the cement stock per capita has reached saturation (Cao et al., 2017) and the concrete structures generally have long service lives (70 and 65 yrs for Europe and the US on average, respectively) relative to the life cycle (i.e. 100 years) considered in our model, it is conceivable that the absolute uptake in these two regions only have been increasing mildly after 1980s, which is in drastic contrast to the 'exponential' rise observed for China (see Figure 6). In terms of ROW, the increase in uptake has been somewhat intermediate between the case for China and developed nations, reflecting their relatively milder increase in cement production/consumption to China yet far from saturation, as well as relatively shorter building service lives to the US and Europe. On the other hand, although cements have been predominantly used for making concretes worldwide, mortar had absorbed more $CO_2$ reaching 12.34 Gt (9.99-14.97 Gt, 95% CI) cumulatively according to our estimates. This is mainly attributed to the faster carbonation kinetics of mortar compared with concrete, manifested by the higher diffusion coefficients, thinner layers of mortar cement and large exposure area in our model as supported by our own experimental measurements (see http://doi.org/10.5281/zenodo.4459729 (Wang et al., 2020)) and literature (Lutz and Bayer, 2010; Winter and Plank, 2007).

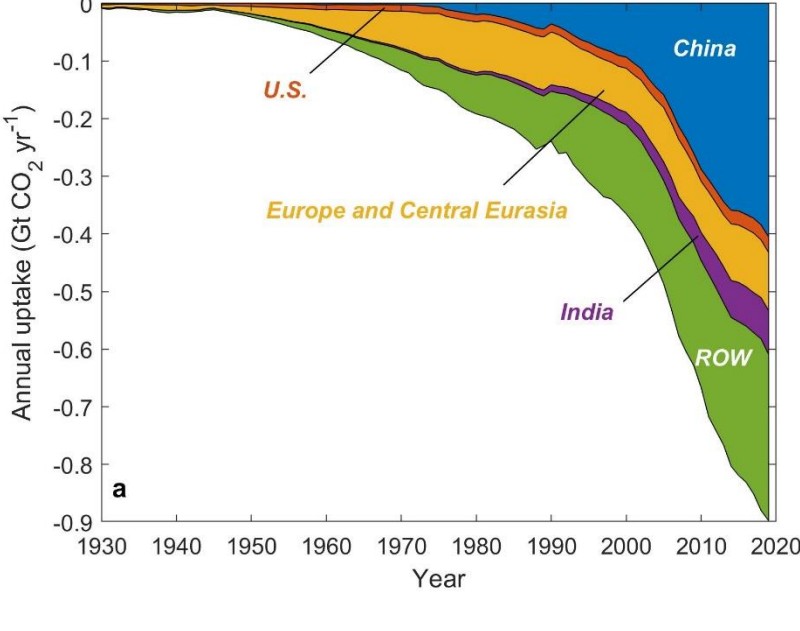

435

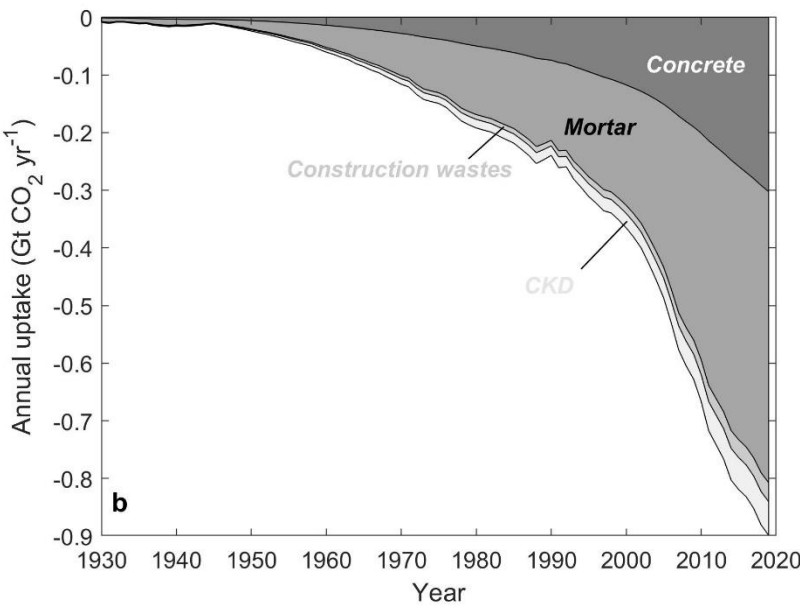

**Figure 5 Annual global uptake (median) by cement materials by a) country/region and b) type from 1930 to 2019. The uptake is projected onto y-axis as negative values, denoting absorption as opposed to emission.**

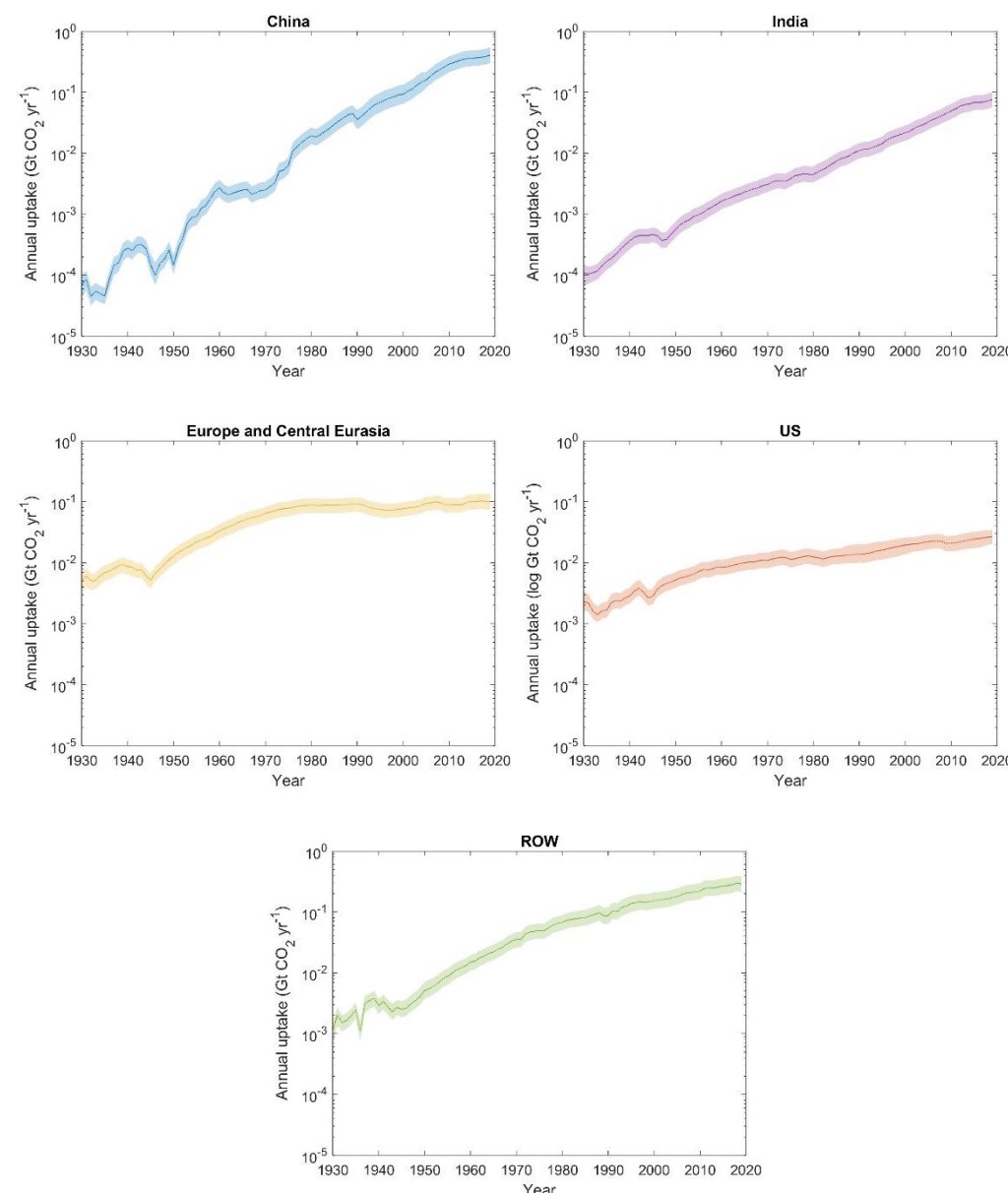

440

**Figure 6 The increase in cement CO₂ uptake in China, India, Europe and Central Eurasia, the US and the rest of the world from 1930 to 2019. The y-axes are plotted in logarithm scale (absolute value) and within the same range for comparison, both the median (dotted lines) and 95% CI (shaded area) are shown.**

445

### 3.3. Characteristics of cement carbon uptake

One of the limitations of natural carbonation for carbon capture is that it is a slow process hence speeding up the chemical processes involved is the key to realise tangible impacts on mitigating $CO_2$ emissions. This is also the case for cement materials especially concrete structures, which took up the majority of their utilisation. Therefore, the carbon uptake by concretes, before demolition, has been persisting during their lifetimes. This is evident in Figure 7 where the cements materials (mainly concretes) consumed in a given decade (colour-coded) still made contribution to carbon uptake decades later. In spite of this feature, more than 71% of the total uptake was attributed to, based on our estimates, the cement materials produced/consumed after the 1990s. This is in line with the trend of process emission growth i.e. 73% for the post-1990 era in the same 1930-2019 period. The difference[14] can be accounted for by the dynamic processes and the varying durations of the stages involved in the life cycles, as considered and implemented in the uptake models. This contrasts with the immediate process emission process. It is also suggested in Figure 7 that a surge in uptake occurs at the demolition stage because of the significant increase in fresh surface area. Figure 8 more evidently demonstrates such an effect by showing the sudden increase of uptake in the late 2000s owing to the concrete produced/consumed historically. This can be traced back to the 1930-1940s when the majority of cement was produced and consumed in Europe, where the average service life of concrete structure is set as 70 years in our model.

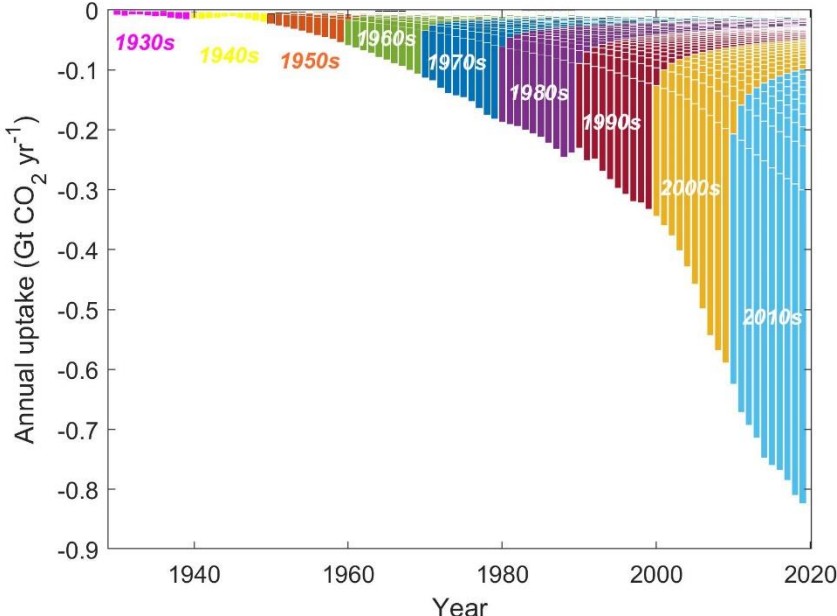

**Figure 7 The cumulative characteristic of carbon uptake of cement. The colour-coded bar areas represent the amount of uptake by the cement produced/consumed in each decade from 1930 to 2019. The fractions of uptake that occurred in each decade post-1990**

---

[14] It is not necessarily the case that the fraction of uptake is smaller than that of the process emission for post-1990 era.

are annotated. The 'tails' indicate that cement produced in a certain time will keep absorbing CO₂ beyond its
production/consumption and the annual uptakes are composed of current and historical contributions.

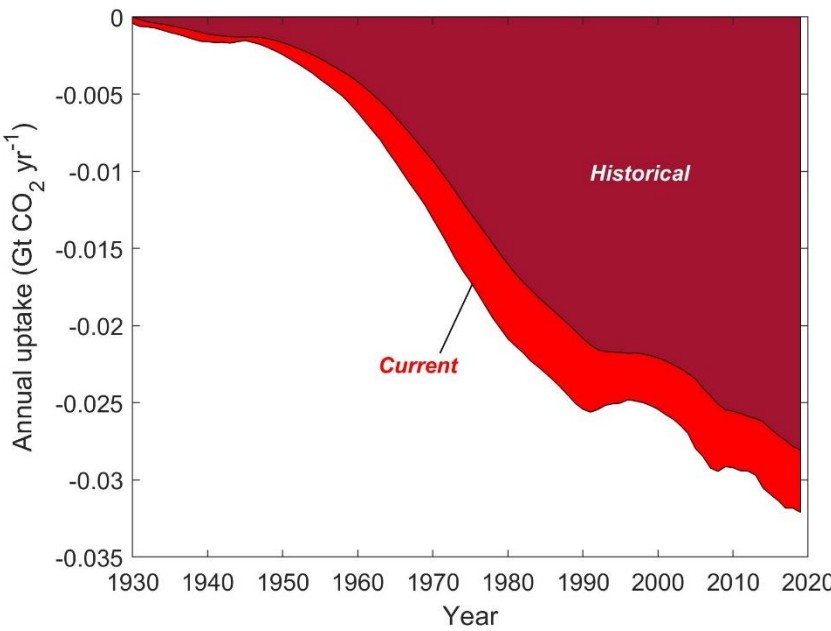

**Figure 8 The annual carbon uptake (median) of concrete produced/consumed in Europe and Central Eurasia, attributed to the current: the concrete produced/consumed in that year and to historical: the concrete produced/consumed in the years prior to that year.**

### 4. Data availability

All the original datasets used for estimating the emission and uptake in this study and the resulting datasets themselves from the simulation, as well as the associated uncertainties are made available by Zenodo at http://doi.org/10.5281/zenodo.4459729 (Wang et al., 2020).

### 5. Conclusions

Estimating $CO_2$ uptake of cements is essential for evaluating the real environmental impact of the cement industry. Previous efforts were limited by data availabilities and incomplete accounting for other cement materials other than concrete. From a historical perspective, while mortar had absorbed more $CO_2$ than any other types of cement, more uptake had occurred in China than in any other country, owing to its dominant cement production/consumption position in recent decades (>43 % from 2000 to 2019). The kinetic processes dictate that $CO_2$ uptake of cement is a dynamic process such that legacy absorption from cements produced in the past should not be omitted. Overall, post-1990 era sees more than 75% of the total uptake

estimated. As a revision to our previous work (Xi et al, 2016) where the clinker ratios were likely to have been overestimated, a dynamic clinker ratio approach was adopted to reflect the recent technological changes in the industry despite limited to China and India only. Besides, the dynamic clinker ratios were also applied in re-evaluating the process emissions. The compounded results suggest that the cumulative $CO_2$ uptake reached 21.02 Gt (18.01-24.41 Gt, 95% CI) offsetting approx. 55% of the corresponding process emission as of 2019. The offset level is noticeably higher than our previous estimate for 1930-2015 (~43%) while the uptake for the same period is broadly similar: 4.8 GtC from this study as opposed to 4.5 GtC from the previous one (Xi et al., 2016), indicating internal consistency of the uptake model and a direct relationship between cement clinker content and process emission.

This dataset and the estimate methodology can serve as a set of tools to assess the emission and, more importantly, the uptake of $CO_2$ by cement materials during their life cycles. Given cement demand is projected to remain increasing to satisfy society developments globally, future work is needed to increase the accuracy of the uptake estimates, crucially, by utilising the direct clinker production data where possible and obtaining spatially resolved conversion factors determined by experiments.

## 6. Author contribution

R.G. prepared, reviewed and edited the manuscript, with assistance from J.Y.W., P.C., S.J.D. and R.M.A.. R.G. performed the analyses, with support from J.Y.W. and L.F.B. on analytical approaches. J.Y.W. curated the datasets. D.T., L.F.B. and F.M.X. developed the code and performed the simulations, with support from R.G., J.Y.W., D.T. and R.M.A. on datasets. F.M.X. and Z.L. conceptualised and supervised the study.

## 7. Competing interests

The authors declare that they have no conflict of interest.

## 8. Acknowledgements

J.Y.W. acknowledges funding from Youth Innovation Promotion Association, Chinese Academy of Sciences (grant no. 2020201). F.M.X. acknowledges funding from the National Natural Science Foundation of China (grant no. 41977290), CAS President's International Fellowship for Visiting Scientists (grant no. 2017VCB0004), Liaoning Xingliao Talents Project (grant no. XLYC1907148), Liaoning Hundred, Thousand and Ten Thousand Talent Project. Z.L. acknowledges funding from the National Natural Science Foundation of China (grant no. 71874097 and 41921005), Beijing Natural Science Foundation (JQ19032) and Qiushi Foundation.

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
