# Peer review of "Global CO2 uptake of cement in 1930-2019"

_Earth System Science Data, 2020_

## Referee Comment (RC1) · Anonymous Referee #1 · 1 Nov 2020

Ms. Ref. No.: essd-2020-275-manuscript-version2 -peer-review-v1 Global CO2 uptake of cement in 1930–2019 Reviewer comments: SUMMARY The manuscript deals with an investigation on the use of an analytical model to estimate the amount of CO2 that had been absorbed from 1930 to 2019 in four types of cement materials including concrete, mortar, construction waste and cement kiln dust (CKD). This is a topic that has not been widely covered in the literature, therefore, this a subject of great interest, but it is somehow limited in the analysis and application of these results. MAIN IMPRESSIONS This paper has an undeniable practical usefulness. However, from a scientific point of view, the following issues must be addressed: i) A key aspect for the IPCC Emission Factor Database is the uncertainties calculation, then, this part should be presented, explained and discussed in detail in the paper; and ii) References and data should be updated.

MORE DETAILED COMMENTS Abstract: Link https://doi.org/10.5281/zenodo.4064803

should not be given in the abstract. Please, include it in the references. Line 32: Please, update the information and references. According to https://doi.org/10.3390/app10010339 "cement production is considered as responsible for approximately 7.4% of the global carbon dioxide emission (2.9 Gtons in 2016)". Line 42: Please, update the information and references. Clinker factor is decreasing according to World Business Council for Sustainable Development (WBCSD); Cement Sustainability Initiative's (CSI). Cement Industry Energy and CO2 Performance. Getting the Numbers Right (GNR) Project, 1st ed.; World Business Council for Sustainable Development: Geneva, Switzerland, 2018. Line 42: " . . . be around 0.5 t CO2/t cement . . . clinker ratio >95% ..." According to Table 3 in reference https://doi.org/10.3390/app10020646, the maximum stoichiometric amount of carbon dioxide that can be absorbed goes from 0.49 kg CO2/kg Cement (for CEM I Portland cement CEM I (OPC)) to 0.10 kg CO2/kg Cement CEM V/B. Line 45: "The universal carbonation mechanisms that are responsible for the carbon uptake of cements can be attributed to their hydroxide(s) and silicate(s) constitutes, as described by Eq. (R1) and (R2):". This is not the only one. In addition, ettringite (https://doi.org/10.1680/adcr.2000.12.3.131) and calcium aluminates may be carbonated at low partial CO2 pressure, resulting in formation of gypsum, alumina gel and vaterite crystals (https://doi.org/10.1016/j.conbuildmat.2011.07.043). Line 45: More precise chemical mechanism that could be referenced are given in https://doi.org/10.1557/JMR.2002.0271, https://doi.org/10.1016/j.cemconcomp.2018.04.006, https://doi.org/10.1680/adcr.2000.12.3.1 Line 49: With regard to this "... multi-giga-tonne potential of CO2 abatement . . .", the effect of the high level of alkaline blending (e.g. blast furnace) for CO2 abatement was proposed previously in ref. https://doi.org/10.3390/en12122346 ("The main finding is the extreme sensitivity of the GGBFS mortars to the curing intensity and, therefore, they can be used cured under controlled conditions to minimize carbon footprints"). Line 50: " ... reducing clinker ratio is still the key to lower the emission level ..." : reducing clinker ratio is one of the key levers to lower the emission level. Could please bring up some

others? For instance, several Carbon Dioxide Uptake levers have been proposed in the Roadmap 2050 of the Cement Industry (Refs.: https://doi.org/10.3390/en13133452 and https://cembureau.eu/media/kuxd32gi/cembureau-2050-roadmap_final-version_web.pdf Line 59: I agree with the statement "... the results by applying more realistic clinker ratio data is necessary ...". Then, I suggest to add the clinker ratios published in the Getting the Numbers Right (GNR) Project, Carbon Capture Technology—Options and Potentials for the Cement Industry, 1st ed.; European Cement Research Academy (ECRA): Düsseldorf, Germany, 2007 and reference https://doi.org/10.3390/app10010339 Line 61: Could you please to cite your paper https://doi.org/10.1038/ngeo2840 ("...has been sequestered in carbonating cement materials from 1930 to 2013, offsetting 43% of the CO2 emissions from production of cement over the same period") and justify this new figure? Line 140: The fraction of CaO that could be converted to CaCO3 is given in Table 3 in reference https://doi.org/10.3390/app10020646. Line 147: The area behind the front cannot be regarded fully carbonated. You should consider a degree of carbonation. Please, check references discussing the effect of the degree of carbonation and surface/volume on the carbonation uptake. Line 187: There are great differences between different countries in the same regional subcategory (Please, check the carbonation behaviour of recycled aggregate concrete reported in https://doi.org/10.1016/j.cemconcomp.2015.04.017. In particular, the use of mineral additions as cement replacement causes greater carbonation depths than those of mixes without them. If you assume a uniform distribution between a and b for each reginal subcategory the uncertainty will be affected. Line 395: "...more than 72% of which have occurred since 1990."" as reported in other papers (See Fig. 9 in ref. https://doi.org/10.3390/en13133452). Line 398: Could you please add a reference? Line 410: Andrew (2018) report could be updated with reference " Robbie M. Andrew. Global CO2 emissions from cement production, 1928–2018. Earth Syst. Sci. Data, 11, 1675–1710, https://doi.org/10.5194/essd-11-1675-2019, 2019"? Line 416: In "3.2. Cement carbon uptake by region and material type", could you please discuss

the main differences between the results given in (Xi et al., 2016) and SI data 4 in "cement carbon emission and uptake results.xlsx". Line 417: Could you please explain in detail how the uncertainty has been calculated? Associated uncertainties are available by Zenodo at https://doi.org/10.5281/zenodo.4064803, however, the excel worksheet (Uncertainty of cement carbon emission and uptake.xlsx) does include any formula. A key aspect for the IPCC Expert Group on Data for the IPCC Emission Factor Database is the uncertainties calculation. Therefore, this part should be highlighted in the paper. Line 429: Could you please add a reference to support "... This is mainly attributed to the faster carbonation kinetics of mortar ...". Line 429: Could you please add a Table with typical diffusion coefficients for mortars and concretes around the world or, at least, provide some references with diffusion coefficients calculated in the main country/regions? Line 240: What about the effect of curing conditions and fly ash, GGBFS, etc., content? Could you please discuss it? (https://doi.org/10.1016/j.cemconcomp.2012.08.024 , https://doi.org/10.1016/j.cemconres.2007.08.014 , https://doi.org/10.1016/j.cemconcomp.2018.04.006 ) Line 440: In Figure 6, which letter corresponds to each area (China, India, the US, Europe and the rest of the world)? Line 447: "... more than 75% of the total uptake was attributed to ... the cement materials produced/consumed after the 1990s ..." as reported in other papers (See Fig. 9 in ref. https://doi.org/10.3390/en13133452). Line 467: Could you please write the equations and procedure used for the simulation, as well as for the associated uncertainties? Line 474: Could you please delete "microscopic". Line 475: In agreement with other papers, it has been found that "post-1990 era sees more than 75% of the total uptake estimated.". Line 477: Could you please give figures about the result of the clinker ratio overestimation? This conclusion should compare clearly the results provided in (Xi et al, 2016) and in the present paper. Line 479: Could you please delete "(see Figure 4a)". Line 480: It is clear that to increase the accuracy of the uptake estimates is necessary. Therefore, conclusions should include the uncertainties obtained in this paper as well as the evaluation of the uncertainty's calculation process. Line 484: Which experiments in

"determined by experiment "?

RECOMMENDATION In conclusion, Major changes have been proposed.

---

## Author Comment (AC1) · 9 Nov 2020

In response to 'SUMMARY and MAIN IMPRESSIONS': Thank you for recognising the importance of this piece of research. Regarding the two issues you mentioned, on the former, we have presented the assumptions, data, statistical distributions and other information in the Monte Carlo simulations, run for calculating the uncertainties in the dataset at https://doi.org/10.5281/zenodo.4064803. On the latter, we will address your concerns in the following texts.

In response to 'MORE DETAILED COMMENTS': Abstract: Including the dataset DOI complies with ESSD guidelines. It is also already included in the references. Line 32: We are happy to update this figure in the revised manuscript. Just to clarify, we simply stressed the share of cement industry emission of the total industrial emission. Line 42: First, this is merely a background introduction of historically high clinker ra-

tios hence emission factors. As part of this work, we explicitly estimated the process emission based on the newest available data. Secondly, the GNR project is known to skew the clinker ratios to the lower end because of its limited coverage. Instead, we used databases of higher resolution and accuracy. Line 42: You are talking about the maximum amount of $CO_2$ that can be absorbed for Portland cement (mainly CEM I).In the manuscript, we referred to two important papers discussing the historically high emission factors of around 0.5 t $CO_2$/t cement. Line 45 (1), (2): Thank you for your advice here. We are aware that the actual carbonation mechanisms are more compli-cated than outlined in the manuscript, and will change the wording to something like 'the main mechanism' or simply add more equations. Nevertheless, the fact that the amount of $CO_2$ that can be taken up depends on the Ca content (active) won't change. Line 49: We don't think what you argued here is in contradiction to what we stated in the manuscript. The '...multi-giga-tonne potential of $CO_2$ abatement...' is partly owing to the low content of clinker due to high mixing. Line 50: We agree with you here. We should have and will stress that reducing clinker ratio is the key to lower the PROCESS emission level. Line 59: As I mentioned above (Line 42), we think the coverage of the GNR project is quite limited hence not representative of the reality. Line 61: This sounds like a good idea. We will add the relevant information in the revised manuscript. Line 140: We considered this parameter and its range explicitly in the Monte Carlo sim-ulations. In the literature you referred to here, this ratio is however fixed at 65%. Line 147: When we say 'fully carbonated', we mean carbonated considering the degree car-bonation e.g. clinker content, active CaO content etc.. 'Fully' here simply suggests that we don't tend to complicate our global-scale model by considering the dynamic evolu-tion of carbonation, like partly-carbonated zone etc. However, we realise that 'fully' is a confusing term and will change the wording accordingly in the revised manuscript. Line 187: Although an assumption, here we applied the uniform distribution between a and b in terms of particle sizes not their carbonation behaviour. Line 395: Can you clarify your question here? As we understand it, the literature you referred to is only concerning Spain using a Tier 1 approach. Line 398: Yes. We will add the dataset at

https://doi.org/10.5281/zenodo.4064803 as the reference as the ratios are listed there. Line 410: Sure. However, since we are only comparing the overlapping 1930-2017 (cumulative) and 2017 (yearly) process emission, either literature would suffice. Line 416: We could but we doing so seems redundant to me the reasoning being we have laid out the differences in the model in line 56,72,367 and 476 between these two studies. The results hence differ accordingly. Therefore, a discussion in this respect would only be comparing numbers with little insights, given the statistical nature of the estimation method. Line 417: We decided not to publish the code at this stage. The uncertainties are estimated using Monte Carlo methods with all the variables, ranges and distributions considered listed in the 'SI table-Variables considered in the uptake uncertainty analysis'. Line 429: This is provided in SI data 9 and SI data 14 tab in 'Input model parameters of cement carbon emission and uptake'. Line 240: Which line are you referring to? Line 440: Noted. They are clearly labelled in the schematics though. Line 447: This is addressed above (Line 395). Line 467: We decided not to publish the code at this stage. Line 474: This seems reasonable, we will do so in the revised manuscript. Line 475: We haven't seen other literature reporting such an index using different models (methods) at global scale. Line 477: Initially we intended to make such a comparison schematically, however, we don't have access to the yearly uptake data as reported in Xi et al. 2016 any more. Line 479: Can you explain the reason? Line 480: It is a good point to include the uncertainties in the conclusion sector, maybe the cumulative results only, given they were already explicitly stated in the preceding sections (Results) and in the SI tables. However, in our opinion, the Conclusion section mainly serve as a summary to catch the trends found in this study. Line 484: this is a proposal for ways to increase the accuracy and reliability of the estimates. Mass spectroscopy and nuclear magnetic resonance, among other experimental methods are useful in determining the conversion factor experimentally.

In response to 'RECOMMENDATION': We believe that the manuscript, together with the dataset provided at https://doi.org/10.5281/zenodo.4064803, is self-contained. We will work on the necessary data update and wording issues the Reviewer pointed out.

Regarding the uncertainty calculation bit, we decided not to publish the code at this stage. The relevant methodologies are in line with Xi et al. (2016) and should be referred to.

---

## Referee Comment (RC2) · Anonymous Referee #1 · 11 Nov 2020

Ms. Ref. No.: essd-2020-275-manuscript-version2 -peer-review-v2 Global CO2 uptake of cement in 1930–2019 Dear Authors, Thank you for your very kind and clear answers.

With regard to your questions:

Line 49: I think that is not in contradiction. It is well-known that cements with low content of clinker lead to lower carbon dioxide footprint. In addition, blast-furnace slag also carbonates as shown in mentioned references.

Line 395: The trends at global and local level scale are similar. Post-1990 period correspond to the highest cement production and, therefore, the highest carbon dioxide uptake. It is suggested to mention other examples or references.

Line 475: Probably in Figure 9 in reference: https://doi.org/10.3390/en13133452 Energies 2020, 13(13), 3452.

[Figure]

Line 479: In the conclusions, references to Figures should be avoided. This is the reason to suggest deleting such reference.

Finally, it is a pity your decision not to publish the uncertainty calculation code for the time being. It would be quite necessary to provide this information in order to include the carbon dioxide uptake in the IPCC Emission Factor Database.

Congratulations for the great work.
* * *

---

## Referee Comment (RC3) · Anonymous Referee #1 · 18 Nov 2020

Thank you for your kin answers. Kind regards!

---

## Author Comment (AC2) · 18 Nov 2020

Thanks again for your kind and swift response. Please see the following for our responses to your suggestions: Line 49: We can agree on this. In addition, cement additives such as blast-furnace slag can accelerate carbonation rate of concrete and mortar (https://doi.org/10.3390/en12122346), this factor has been explicitly considered in our study (see the SI data 9 in the 'Input model parameters of cement carbon emission and uptake' file). Meanwhile, calcium oxide in cement additives also carbonates (https://doi.org/10.3390/en12122346). However, in order to meet the performance standards for cement materials, the CaO content usually does not change noticeably. In our study, we took this aspect of uncertainty into account as well, hence did not use the constant value. Line 395: In the revised manuscript, we will add necessary comparative analysis. The paper by Cao et al. (2020) (doi: 10.1038/s41467-020-17583-w) is a proper candidate. The literature you referred to is only concerning Spain using

a simple transformation approach according to IPCC Guidelines (ACDU (service life) = $\alpha \times$IPCC reported emissions due to the calcination process; ACDU (end-of-life) = $\beta \times$IPCC reported emissions due to the calcination process, with $\alpha$ and $\beta$ being 0.20 and 0.03, respectively), which is totally different to our cement uptake models. There is little comparability between them. Line 475: The same as Line 395. Line 479: This seems reasonable. Thanks. Regarding the uncertainty calculation bit, thank you for your advice here. We are aware that providing the uncertainty calculation code is necessary for our results to be included in the IPCC Emission Factor database. At this stage, however, we are still in the process of copyrighting the code thus decided not to publish the code, yet. Thank you for recognising the importance of this piece of research.

---

## Referee Comment (RC4) · Anonymous Referee #2 · 17 Jan 2021

General comments This manuscript works on an investigation on the use of an analytical model to estimate the amount of CO2 uptake from 1930 to 2019 in four types of cement materials including concrete, mortar, construction waste and cement kiln dust. It is a topic that has not been widely covered in the literature, and therefore, a subject of great interest, but it is somehow limited in the analysis and application of these results. This paper is useful for evaluating the real environmental impact of the cement industry. This dataset and the estimate methodology may serve as a set of tools to assess the emission and, more importantly, the uptake of CO2 by cement materials during their life cycles.

Specific comments Carbonation of cement produces calcite, whose dissolution also consume CO2. How do you consider this effect of calcite dissolution on the CO2 uptake of cement?

---

## Author Response (AR1)

**RC1**

General comments: 'SUMMARY The manuscript deals with an investigation on the use of an analytical model to estimate the amount of CO2 that had been absorbed from 1930 to 2019 in four types of cement materials including concrete, mortar, construction waste and cement kiln dust (CKD). This is a topic that has not been widely covered in the literature, therefore, this a subject of great interest, but it is somehow limited in the analysis and application of these results. MAIN IMPRESSIONS This paper has an undeniable practical usefulness. However, from a scientific point of view, the following issues must be addressed: i) A key aspect for the IPCC Emission Factor Database is the uncertainties calculation, then, this part should be presented, explained and discussed in detail in the paper; and ii) References and data should be updated.'

Response: Thank you for recognising the importance of this piece of research. Regarding the two issues you mentioned, on the former, we have presented the assumptions, data, statistical distributions and other information in the Monte Carlo simulations, run for calculating the uncertainties in the dataset at https://doi.org/10.5281/zenodo.4064803. On the latter, we will address your concerns in the following texts.

Changes: None.

1. Comment: 'Abstract: Link https://doi.org/10.5281/zenodo.4064803 should not be given in the abstract. Please, include it in the references.'

Response: Including the dataset DOI complies with ESSD guidelines. It is also already included in the references.

Changes: None.

2. Comment: 'Line 32: Please, update the information and references. According to https://doi.org/10.3390/app10010339 "cement production is considered as responsible for approximately 7.4% of the global carbon dioxide emission (2.9 Gtons in 2016)".'

Response: We are happy to update this figure in the revised manuscript. Just to clarify, we simply stressed the share of cement industry emission of the total industrial emission.

Changes: Adding 'and estimated to account for approximately 7.4% of the total anthropogenic $CO_2$ emissions in 2016' and the corresponding reference.

3. Comment: 'Line 42: Please, update the information and references. Clinker factor is decreasing according to World Business Council for Sustainable Development (WBCSD); Cement Sustainability Initiative's (CSI). Cement Industry Energy and CO2 Performance. Getting the Numbers Right (GNR) Project, 1st ed.; World Business Council for Sustainable Development: Geneva, Switzerland, 2018.'

Response: First, this is merely a background introduction of historically high clinker ratios hence emission factors. As part of this work, we explicitly estimated the process emission based on the newest available data. Secondly, the GNR project is known to skew the clinker ratios to the lower end because of its limited coverage. Instead, we used databases of higher resolution and accuracy.

Changes: None.

4. Comment: 'Line 42: " . . . be around 0.5 t CO2/t cement . . . clinker ratio >95% ..." According to Table 3 in reference https://doi.org/10.3390/app10020646, the maximum stoichiometric amount of carbon dioxide that can be absorbed goes from 0.49 kg CO2/kg Cement (for CEM I Portland cement CEM I (OPC)) to 0.10 kg CO2/kg Cement CEM V/B.'

Response: You are talking about the maximum amount of $CO_2$ that can be absorbed for Portland cement (mainly CEM I). In the manuscript, we referred to two important papers discussing the historically high emission factors of around 0.5 t $CO_2$/t cement.

Changes: None.

5. Comment: 'Line 45: "The universal carbonation mechanisms that are responsible for the carbon uptake of cements can be attributed to their hydroxide(s) and silicate(s) constitutes, as described by Eq. (R1) and (R2):". This is not the only one. In addition, ettringite (https://doi.org/10.1680/adcr.2000.12.3.131) and calcium aluminates may be carbonated at low partial CO2 pressure, resulting in formation of gypsum, alumina gel and vaterite crystals (https://doi.org/10.1016/j.conbuildmat.2011.07.043). Line 45: More precise chemical mechanism that could be referenced are given in https://doi.org/10.1557/JMR.2002.0271, https://doi.org/10.1016/j.cemconcomp.2018.04.006, https://doi.org/10.1680/adcr.2000.12.3.131.'

Response: We are aware that the actual carbonation mechanisms are more complicated than outlined in the manuscript. However, quantitatively carbonation of cement (e.g. OPC) is mainly attributed to CH and C-S-H phases (after hydration). We will change the wording to something like 'the main mechanism' and mention more

other carbonation reactions. Nevertheless, the fact that the amount of $CO_2$ that can be taken up depends on the active Ca content, which is mainly derived from lime, will not change.

Changes: Replacing 'universal' with 'main'. Adding a footnote mentioning carbonation of other phases and the corresponding reference.

6. Comment: 'Line 49: With regard to this "... multi-giga-tonne potential of CO2 abatement . . .", the effect of the high level of alkaline blending (e.g. blast furnace) for CO2 abatement was proposed previously in ref. https://doi.org/10.3390/en12122346 ("The main finding is the extreme sensitivity of the GGBFS mortars to the curing intensity and, therefore, they can be used cured under controlled conditions to minimize carbon footprints").'

Response: We don't think what you argued here is in contradiction to what we stated in the manuscript. The '...multi-giga-tonne potential of $CO_2$ abatement...' is partly owing to the low content of clinker due to high mixing.

Changes: None.

7. Comment: 'Line 50: " ... reducing clinker ratio is still the key to lower the emission level ..." : reducing clinker ratio is one of the key levers to lower the emission level. Could please bring up some others? For instance, several Carbon Dioxide Uptake levers have been proposed in the Roadmap 2050 of the Cement Industry (Refs.: https://doi.org/10.3390/en13133452 and https://cembureau.eu/media/kuxd32gi/cembureau-2050-roadmap_finalversion_web.pdf'

Response: We agree with you here. We should have and will stress that reducing clinker ratio is the key to lower the PROCESS emission level.

Changes: Adding 'process' in this sentence.

8. Comment: 'Line 59: I agree with the statement ". . . the results by applying more realistic clinker ratio data is necessary . . .". Then, I suggest to add the clinker ratios published in the Getting the Numbers Right (GNR) Project, Carbon Capture TechnologyˆA˘TOptions and Potentials for the Cement Industry, 1st ed.; ˘ European Cement Research Academy (ECRA): Düsseldorf, Germany, 2007 and reference https://doi.org/10.3390/app10010339'.

Response: As I mentioned above regarding Line 42 comment, we think that the coverage of the GNR project is quite limited hence not representative of the reality.

Changes: None.

9. Comment: 'Line 61: Could you please to cite your paper https://doi.org/10.1038/ngeo2840 ("...has been sequestered in carbonating cement materials from 1930 to 2013, offsetting 43% of the CO2 emissions from production of cement over the same period") and justify this new figure?'

Response: Line 61: This sounds like a good idea. However, this justification might appear too early in the context of the structure of the manuscript. Instead, we will add the relevant information in the Conclusion section in the revised manuscript.

Changes: Adding 'The offset level is noticeably higher than our previous estimate for 1930-2015 (~43%) while the uptake for the same period is broadly similar: 4.8 GtC from this study as opposed to 4.5 GtC from the previous one (Xi et al., 2016), indicating internal consistency of the uptake model and a direct relationship between cement clinker content and process emission.', at the end of the first paragraph of Conclusion section.

10. Comment: 'Line 140: The fraction of CaO that could be converted to CaCO3 is given in Table 3 in reference https://doi.org/10.3390/app10020646.'

Response: Line 140: We considered this parameter and its range explicitly in the Monte Carlo simulations. In the literature you referred to here, this ratio is however fixed at 65%.

Changes: None.

11. Comment: 'Line 147: The area behind the front cannot be regarded fully carbonated. You should consider a degree of carbonation. Please, check references discussing the effect of the degree of carbonation and surface/volume on the carbonation uptake.'

Response: When we say 'fully carbonated', we mean carbonated considering the degree carbonation e.g. clinker content, active CaO content etc.. 'Fully' here simply suggests that we don't tend to complicate our global-scale model by considering the dynamic evolution of carbonation, like partly-carbonated zone etc. However, we realise that 'fully' is a confusing term and will change the wording accordingly in the revised manuscript.

Changes: Adding a footnote saying 'As opposed to the concept 'partly-carbonated', where reaction kinetics are considered.'

12. Comment: 'Line 187: There are great differences between different countries in the same regional subcategory (Please, check the carbonation behaviour of recycled

aggregate concrete reported in https://doi.org/10.1016/j.cemconcomp.2015.04.017. In particular, the use of mineral additions as cement replacement causes greater carbonation depths than those of mixes without them. If you assume a uniform distribution between a and b for each reginal subcategory the uncertainty will be affected.'

Response: Despite being an assumption, here we applied the uniform distribution between a and b in terms of particle sizes not their carbonation behaviour.

Changes: None.

13. Comment: 'Line 395: ". . .more than 72% of which have occurred since 1990.,,," as reported in other papers (See Fig. 9 in ref. https://doi.org/10.3390/en13133452).'

Response: Can you clarify your question here? As we understand it, the literature you referred to is only concerning Spain using a Tier 1 approach.

Changes: None.

14. Comment: 'Line 398: Could you please add a reference?'

Response: Yes. We will add the reference of our submitted dataset Zenodo.

Changes: Adding the reference '(Wang et al., 2020)'.

15. Comment: 'Line 410: Andrew (2018) report could be updated with reference " Robbie M. Andrew. Global CO2 emissions from cement production, 1928–2018. Earth Syst. Sci. Data, 11, 1675–1710, https://doi.org/10.5194/essd-11-1675-2019, 2019"?'

Response: Sure. However, since we are only comparing the overlapping 1930-2017 (cumulative) and 2017 (yearly) process emission, either literature would suffice.

Changes: Changing the reference to 'Andrew (2019)' and updating in the reference list.

16. Comment: 'Line 416: In "3.2. Cement carbon uptake by region and material type", could you please discuss the main differences between the results given in (Xi et al., 2016) and SI data 4 in "cement carbon emission and uptake results.xlsx".'

Response: We could but doing so seems redundant to me with the reasoning being that we have laid out the differences in the model in line 56,72,367 and 476 (before the changes) between these two studies. The results hence differ accordingly. Therefore, a discussion in this respect would only be serving to compare numbers with little insight, given the statistical nature of the estimation method.

Changes: None.

17. Comment: 'Line 417: Could you please explain in detail how the uncertainty has been calculated? Associated uncertainties are available by Zenodo at https://doi.org/10.5281/zenodo.4064803, however, the excel worksheet (Uncertainty of cement carbon emission and uptake.xlsx) does include any formula. A key aspect for the IPCC Expert Group on Data for the IPCC Emission Factor Database is the uncertainties calculation. Therefore, this part should be highlighted in the paper.'

Response: We decided not to publish the code at this stage. The uncertainties are estimated using Monte Carlo methods with all the variables, ranges and distributions considered listed in the 'SI table-Variables considered in the uptake uncertainty analysis'.

Changes: None.

18. Comment: 'Line 429: Could you please add a reference to support ". . . This is mainly attributed to the faster carbonation kinetics of mortar . . .". Line 429: Could you please add a Table with typical diffusion coefficients for mortars and concretes around the world or, at least, provide some references with diffusion coefficients calculated in the main country/regions?'

Response: This is provided in SI data 9 and SI data 14 tab in 'Input model parameters of cement carbon emission and uptake'. We have also further guided the reader to where to look for such evidence in the following lines.

Changes: None.

19. Comment: 'Line 240: What about the effect of curing conditions and fly ash, GGBFS, etc., content? Could you please discuss it?
(https://doi.org/10.1016/j.cemconcomp.2012.08.024 , https://doi.org/10.1016/j.cemconres.2007.08.014 ,
https://doi.org/10.1016/j.cemconcomp.2018.04.006 )'

Response: Which line are you referring to? It seems there is a mix-up.

Changes: None.

20. Comment: 'Line 440: In Figure 6, which letter corresponds to each area (China, India, the US, Europe and the rest of the world)?'

Response: Noted. We will change the format of the in the figures in the revised manuscript. They are clearly labelled in the schematics though.

Changes: See Figure 6 and its caption.

21. Comment: 'Line 447: ". . . more than 75% of the total uptake was attributed to . . . the cement materials produced/consumed after the 1990s . . ." as reported in other papers (See Fig. 9 in ref. https://doi.org/10.3390/en13133452).'

Response: This is addressed above for the 'Line 395' comment.

Changes: None.

22. Comment: 'Line 467: Could you please write the equations and procedure used for the simulation, as well as for the associated uncertainties?'

Response: Again, we decided not to publish the code at this stage while we are applying for patents.

Changes: None.

23. Comment: 'Line 474: Could you please delete "microscopic".'

Response: This seems reasonable, we will do so in the revised manuscript.

Changes: Deleting 'microscopic'.

24. Comment: 'Line 475: In agreement with other papers, it has been found that "post-1990 era sees more than 75% of the total uptake estimated.".'

Response: We have not seen other literature reporting such an index using different models (methods) at global scale.

Changes: None.

25. Comment: 'Line 477: Could you please give figures about the result of the clinker ratio overestimation? This conclusion should compare clearly the results provided in (Xi et al, 2016) and in the present paper.'

Response: Initially we intended to make such a comparison schematically, however, we don't have access to the yearly uptake data as reported in Xi et al. 2016 any more.

Changes: None.

26. Comment: 'Line 479: Could you please delete "(see Figure 4a)".'

Response: Can you explain why?

Changes: None.

27. Comment: 'Line 480: It is clear that to increase the accuracy of the uptake estimates is necessary. Therefore, conclusions should include the uncertainties

obtained in this paper as well as the evaluation of the uncertainty's calculation process.'

Response: It is a good point to include the uncertainties in the conclusion sector, but maybe the cumulative results only, given the others were already explicitly stated in the preceding sections (Results) and in the SI tables. In our opinion, the Conclusion section mainly serve as a summary to catch the trends found in this study.

Changes: Changing the sentence 'The compounded results suggest that the cumulative CO2 offset reached approx. 52% as of 2019' to 'The compounded results suggest that the cumulative $CO_2$ uptake reached 21.12 Gt (18.12-24.54 Gt, 95% CI) offsetting approx. 52% of the corresponding process emission as of 2019'.

28. Comment: 'Line 484: Which experiments in "determined by experiment "?'

Response: This is a proposal for ways to increase the accuracy and reliability of the estimates. Mass spectroscopy and nuclear magnetic resonance, among other experimental methods are useful in determining the conversion factor experimentally.

Changes: None.

**RC2**

1. Comment: 'Line 49: I think that is not in contradiction. It is well-known that cements with low content of clinker lead to lower carbon dioxide footprint. In addition, blast-furnace slag also carbonates as shown in mentioned references.'

Response: We can agree on this. In addition, cement additives such as blast-furnace slag can accelerate carbonation rate of concrete and mortar (https://doi.org/10.3390/en12122346), this factor has been explicitly considered in our study (see the SI data 9 in the 'Input model parameters of cement carbon emission and uptake' file). Meanwhile, calcium oxide in cement additives also carbonates (https://doi.org/10.3390/en12122346). However, in order to meet the performance standards for cement materials, the CaO content usually does not change noticeably. In our study, we took this aspect of uncertainty into account as well, hence did not use the constant value.

Changes: None.

2. Comment: 'Line 395: The trends at global and local level scale are similar. Post-1990 period correspond to the highest cement production and, therefore, the highest carbon dioxide uptake. It is suggested to mention other examples or references.'

Response: In the revised manuscript, we will add necessary comparative analysis. The paper by Cao et al. (2020) (doi: 10.1038/s41467-020-17583-w) is a proper

candidate. The literature you referred to is only concerning Spain using a simple transformation approach according to IPCC Guidelines (ACDU (service life) = $\alpha \times$IPCC reported emissions due to the calcination process; ACDU (end-of-life) = $\beta \times$IPCC reported emissions due to the calcination process, with $\alpha$ and $\beta$ being 0.20 and 0.03, respectively), which is totally different to our cement uptake models. There is little comparability between them.

Changes: Adding 'This finding agrees with other studies on cement carbon uptake using similar modelling approaches (Cao et al., 2020)' and updating the reference list accordingly.

3. Comment: 'Line 475: Probably in Figure 9 in reference: https://doi.org/10.3390/en13133452 Energies 2020, 13(13), 3452.'

Response: The same as the response above (Line 395).

Changes: None.

4. Comment: 'Line 479: In the conclusions, references to Figures should be avoided. This is the reason to suggest deleting such reference.'

Response: This seems reasonable.

Changes: Deleting '(see Figure 4a).

5. Comment: 'Finally, it is a pity your decision not to publish the uncertainty calculation code for the time being. It would be quite necessary to provide this information in order to include the carbon dioxide uptake in the IPCC Emission Factor Database.'

Response: We are aware that providing the uncertainty calculation code is necessary for our results to be included in the IPCC Emission Factor database. At this stage, however, we are still in the process of copyrighting the code thus decided not to publish the code, yet.

Changes: None.

**RC3**

Not applicable

**RC4**

General comments: 'This manuscript works on an investigation on the use of an analytical model to estimate the amount of CO2 uptake from 1930 to 2019 in four

types of cement materials including concrete, mortar, construction waste and cement kiln dust. It is a topic that has not been widely covered in the literature, and therefore, a subject of great interest, but it is somehow limited in the analysis and application of these results. This paper is useful for evaluating the real environmental impact of the cement industry. This dataset and the estimate methodology may serve as a set of tools to assess the emission and, more importantly, the uptake of $CO_2$ by cement materials during their life cycles.'

Response: Thank you for recognizing the importance of this piece of research. In this article, we focused on updating the global cement carbon uptake inventory and its distribution, the detailed analyses such as how the carbonation factor affects the uptake had been presented in our previous work (doi: 10.1038/NGEO2840), Therefore, we did not place special emphasis on the analysis of the results. Our results demonstrate that carbonation of cement products is an important anthropogenic carbon sink, which has not been thoroughly assessed or documented. Using our consistent framework and model, regular updating the annual and cumulative estimates of cement carbon uptake can be realized, so that their inclusion in the global carbon budget is foreseen. Additionally, our work can bring instruction and inspiration for carbon capture technology and carbon neutralisation path.

Changes: None.

1. Comment: 'Carbonation of cement produces calcite, whose dissolution also consume $CO_2$. How do you consider this effect of calcite dissolution on the $CO_2$ uptake of cement?'

Response: Cement carbonation produces calcite by aqueous precipitation reactions, and it is the main and the most stable polymorph. While calcite dissolution does take place in nature e.g., prominently in the karst area, in typical micro-environments of cement/concrete, calcite dissolution reactions are not favoured because calcite is continuously supersaturated to enhance precipitation.
(https://doi.org/10.1016/j.jcou.2020.02.015;
https://doi.org/10.1016/j.jcou.2020.101428). In addition, other co-existing phases such as calcium hydroxide have much higher solubility than calcite. Existing research have shown that in the karst area dissolving 1 mole of calcium carbonate consumes 1 mole of $CO_2$. Based on this theory, calcite dissolution could be helpful to the $CO_2$ uptake of cement. However, it is not the main chemical mechanism in cement materials life cycles. Hence, we did not consider the effect of calcite dissolution on the $CO_2$ uptake of cement. Nevertheless, quantitative determination of calcite

dissolution in cement products and its effects may be an important research topic in light of global warming and acid atmospheric deposition.

Changes: None.